# Detection of Image Steganography Using Deep Learning and Ensemble Classifiers

**Mikołaj Płachta** * , **Marek Krzemień** , **Krzysztof Szczypiorski** and **Artur Janicki** *

Faculty of Electronics and Information Technology, Warsaw University of Technology, Nowowiejska 15/19, 00-665 Warsaw, Poland; marek.krzemien.stud@pw.edu.pl (M.K.); krzysztof.szczypiorski@pw.edu.pl (K.S.)
* Correspondence: mikolaj.plachta.dokt@pw.edu.pl (M.P.); artur.janicki@pw.edu.pl (A.J.)

**Abstract:** In this article, the problem of detecting JPEG images, which have been steganographically manipulated, is discussed. The performance of employing various shallow and deep learning algorithms in image steganography detection is analyzed. The data, images from the BOSS database, were used with information hidden using three popular steganographic algorithms: JPEG universal wavelet relative distortion (J-Uniward), nsF5, and uniform embedding revisited distortion (UERD) at two density levels. Various feature spaces were verified, with the discrete cosine transform residuals (DCTR) and the Gabor filter residuals (GFR) yielding best results. Almost perfect detection was achieved for the nsF5 algorithm at 0.4 bpnzac density (99.9% accuracy), while the detection of J-Uniward at 0.1 bpnzac density turned out to be hardly possible (max. 56.3% accuracy). The ensemble classifiers turned out to be an encouraging alternative to deep learning-based detection methods.

**Keywords:** steganography; machine learning; image processing; BOSS database; ensemble classifier; deep learning; steganalysis; stegomalware





## 1. Introduction

Steganography is a method of hiding classified information in non-secret material. In other words, we can hide a secret message in data that we publicly send or deliver, hiding the very existence of a secret communication. Steganographic methods pose a significant threat to users, as they can be used to spread malicious software, or can be used by such malware (so-called stegomalware [1]), for example, for C&C communications or to leak sensitive data.

An important share of steganographic methods use multimedial data, including images, as a carrier. These methods are often referred to as digital media steganography and image steganography, respectively. An example is a method used by the Vawtrak/Neverquest malware [2], the idea of which was to hide URL addresses within favicon images. Another example would be the Invoke-PSImage [3] tool, where developers hid PowerShell scripts in image pixels using a commonly used least-significant bit (LSB) approach. Yet another variance may be hiding information in the structure of GIF files [4], which is quite innovative due to the binary complexity of the GIF structure.

It is observed that a growing number of malware infections take advantage of some kinds of hidden transmission, including that based on image steganography. Since malware infections pose a significant threat to the security of users worldwide, finding efficient, reliable, and fast methods of detecting hidden content becomes very important. Therefore, numerous initiatives and projects have been recently initiated to increase malware and stegomalware resilience–one of them is the Secure Intelligent Methods for Advanced RecoGnition of malware and stegomalware (SIMARGL) project [5], realized within the EU Horizon 2020 framework.

The experiments presented in this article are part of this initiative. The aim of our research was to find the most effective automatic methods for detecting digital steganography in JPEG images. JPEG-compressed images are usually stored in files with extensions:

.jpeg, .jpg, .jpe, .jif, .jfif, and .jfi. JPEG compression is commonly used for image storage and transfer; according to [6], 74.3% of web pages contain JPEG images. Therefore, these images can also be easily used for malicious purposes. In this study, we researched various machine learning (ML) methods of creating predictive models able to discover steganographically hidden content that can be potentially used by malware. Such a detection method can be integrated with antimalware software or any other system performing file scanning for security purposes (e.g., a messaging system).

The great advantage of our research is that we experimented both with shallow ML algorithms and with deep learning methods. As for shallow algorithms, we focused on ensemble classifiers, which have been recently shown to yield good results in detection tasks. When dealing with deep learning methods, we concentrated on a lightweight approach, which did not involve computationally-intensive convolutional layers in a neural network architecture. However, for the sake of simplicity, we did not research the impact of hidden content on detection accuracy—all experiments were conducted with random hidden messages.

Our article is structured as follows: first, in Section 2, we briefly review the state of the art in the area of hiding data in digital images and its detection. Next, in Section 3, we describe the experimental environment, including the test scenarios and the evaluation metrics used. The results are described in Section 4. The article concludes with discussion of the results in Section 5 and a summary in Section 6.

## 2. Related Work

This article focuses on JPEG images as carriers of steganographically embedded data. The popularity of this file format has resulted in a number of data-hiding methods being proposed, as well as various detection methods. In this section, we briefly review the basics of JPEG-based image steganography, including the most commonly used algorithms. Next, we proceed to the detection methods.

### 2.1. JPEG-Based Image Steganography

While multiple steganographic algorithms operate in the spatial domain, there are some that introduce changes on the level of discrete cosine transform (DCT) coefficients stored in JPEG files. Moreover, certain algorithms are designed to minimize the probability of detection through the use of content-adaptiveness: they embed data predominately in less predictable regions, where changes are more difficult to identify. Such modifications are the most challenging to detect; this is why we selected them for our study. Following other studies, e.g., [7], we chose nsF5 [8], JPEG universal wavelet relative distortion (J-Uniward) [9], and uniform embedding revisited distortion (UERD) [10]. They are briefly characterized in the following subsections.

#### 2.1.1. nsF5

The nsF5 [8] algorithm embeds data by modifying the least significant bits of AC ("alternating current", having at least one non-zero frequency) DCT coefficients of JPEG cover objects. Data is hidden using syndrome coding. Assuming that the sender has a $p$-bit message $m \in \{0,1\}^p$ to embed using $n$ AC DCT values with their least significant bits $x \in \{0,1\}^n$ while only $k$ coefficients $x_i$, $i \in I$ are non-zero, only some bits $x_i$, $i \in I$ are modified, thus receiving $y \in \{0,1\}^n$. This vector needs to satisfy:

$$Dy = m,$$

where $D$ is a binary $p \times n$ matrix that is shared between the sending and receiving party. The embedding party needs to find the solution for the aforementioned equation that does not require modifying the bits of zero-valued coefficients ($x_i = y_i$, $i \notin I$). The solution needs to minimize the Hamming weight between the modified and unmodified least-significant-bit vectors ($x - y$). Using this coding method allows the sender to introduce fewer changes than there are bits to embed, thus decreasing the impact of embedding

on the carrier object. While the example provided shows how syndrome coding works, usually a more sophisticated coding scheme, syndrome trellis coding (STC) [11], using a parity-check matrix in place of $D$, is applied. The y vector represents a path through a trellis built based on the parity-check matrix.

### 2.1.2. J-Uniward

J-Uniward [9] is a method for modeling steganographic distortion caused by data embedding. It aims to provide a function that determines which regions of the cover object are less predictable and harder to model. Changes introduced during steganographic data embedding in those areas are harder to detect than if they were introduced uniformly across the carrier. Through computation of relative changes of values based on directional filter bank decomposition this method is able to detect smooth edges that are easy to model. By detecting these predictable and unpredictable areas, this method provides a way of determining where embedding changes would be least noticeable. This method is paired with a coding scheme, such as syndrome trellis coding (STC), to create a content-adaptive data-hiding algorithm.

### 2.1.3. UERD

UERD [10] is a steganographic embedding scheme that aims to minimize the probability of steganographically encoded information's presence being detected, by minimizing the embedding's impact on the statistical parameters of the cover information. It achieves this by analyzing the parameters of DCT coefficients of given modes, as well as whole DCT blocks and their neighbors. Through this, the method can determine whether the region can be considered "noisy" and whether embedding will impact statistical features such as histograms of the file. "Wet" regions are those where statistical parameters are predictable and where embedding would cause noticeable changes. The scheme does not exclude values such as the DC mode coefficients or zero DCT coefficients from being used when embedding, as their statistical profiles can make them suitable from the security perspective. UERD attempts to uniformly spread the relative changes of statistics resulting from embedding. UERD employs syndrome trellis coding (STC) to hide message bits in the desired values.

Figure 1 shows a sample clean image, the same image with random data hidden using the UERD algorithm at 0.4 bpnzac (bits per non-zero AC DCT coefficient) rate, and an image which is the difference between them. As can be observed, despite there being almost 5% hidden data in the image (b) no artifacts can be perceived. What is more, it is hardly possible to observe any difference between the clean and steganographically-modified image, even if they are displayed next to one another. It is only the differential image (c) that proves the manipulation. The same refers to nsF5, J-Uniward, and other modern algorithms realizing image steganography—their manipulations are often imperceptible and difficult to detect, considering that the original image is rarely available.

### 2.2. Detection Methods

In recent years, several methods of detecting image steganography have been researched. They usually involve the extraction of some sort of parameters out of analyzed images, followed by applying a classification algorithm. They are usually based on an ML approach, employing either shallow or deep learning algorithms. Therefore, in this subsection, we first describe the features most frequently used with steganalytic algorithms, and then briefly describe typical examples of shallow and deep learning-based detection algorithms.

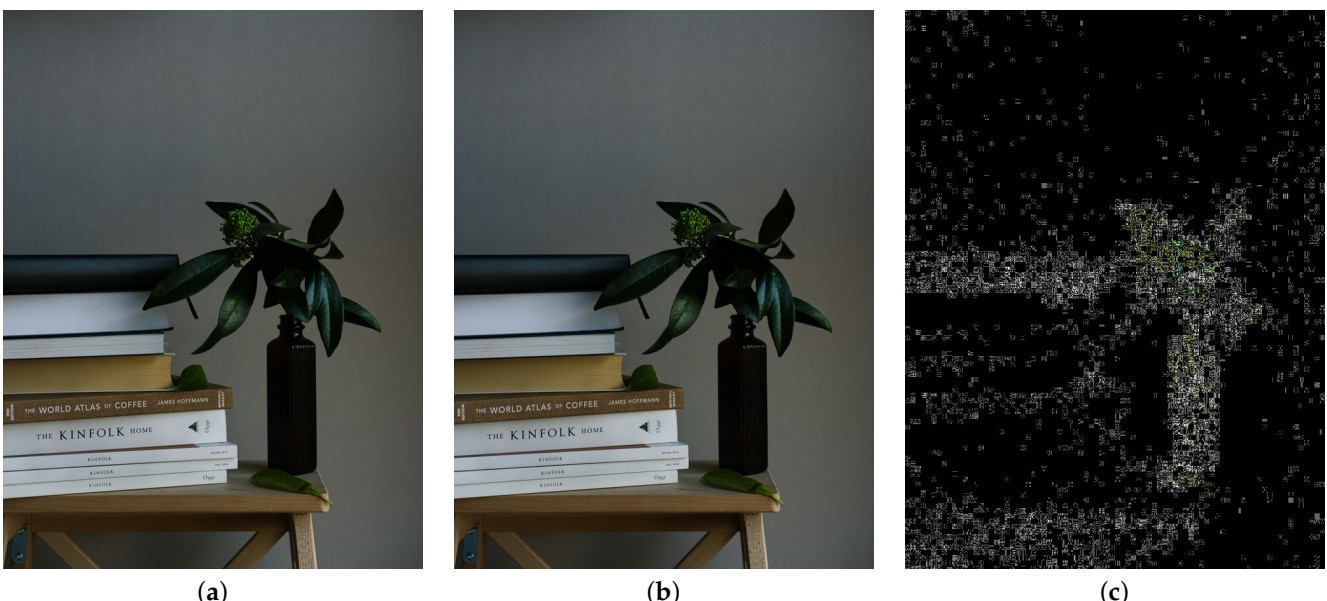

<div align="center">(<b>a</b>)        (<b>b</b>)        (<b>c</b>)</div>

**Figure 1.** (**a**) Clean image, (**b**) image with data hidden using UERD algorithm, and (**c**) differential image between them, scaled 100 times. Density of steganographic data: 0.4 bpnzac, which means here 2638 B of hidden data in each 53 kB image file. Clean image source: unsplash.com (accessed on 8 April 2022).

### 2.2.1. Feature Extraction

In the literature, several feature spaces for image steganalysis have been researched. One of them is based on discrete cosine transform residuals (DCTR) [12], the main purpose of which is to analyze the data resulting from obtaining the DCT value for a given image. First, in this method, a random 8 × 8 pixel filter is created that will be applied to the entire analyzed set. Then, a histogram is created after applying the convolution function with the previously mentioned filter, iterating through each fragment of the analyzed image. In [13], an example of using DCTR parameters in connection with a multi-level filter is proposed. A different variation of this approach is a method based on gabor filter residuals (GFR) [14]. It works in a very similar way to DCTR, but instead of a random 8 × 8 filter, Gabor filters are used. Article [15] describes a successful application of GFR features in JPEG steganography detection. Another approach to parameterization is using the phase aware projection model (PHARM) [16]. In this approach, various linear and non-linear filters are used, while the histogram is constructed from the projection of values for each residual image fragment.

### 2.2.2. Shallow Machine Learning Classifiers

A number of shallow classification methods have been proposed for JPEG steganalysis. These include the use of algorithms such as support vector machines (SVM) [17–19] or logistic regression [20]. A method often appearing in recent publications is an ensemble classifier built using the Fisher linear discriminant (FLD) as the base learner [21]. In certain cases [7], parameter extractors coupled with this ensemble classifier outperformed more recent deep learning-based systems. As such, this algorithm has become a point of reference when looking into the performance of shallow ML methods in detecting steganography. The rationale driving attempts to increase its detection accuracy is the fact that data is split randomly into subsets used to train each base learner. Thus, it may be possible that certain base learners are assigned less varied datasets. Their detection accuracy may suffer from poor generalization capabilities. Simple ensemble vote-combining methods such as the one used by default do not take such effects into consideration.

### 2.2.3. Deep Learning Methods

In recent years, neural networks have often been reported as being used for detecting steganographically hidden data in digital images. As input data, extracted image parameters based on decompressed DCT values such as DCTR, GFR, or PHARM have been used. Proprietary variants of convolutional networks such as XuNet [22], ResNet [23], DenseNet [24], or AleksNet [25] are most often used for this purpose. The common feature of these networks is combining the convolution-batch normalization-dense structures, i.e., the convolutional function, the normalization layer, and the basal layer of neurons with the appropriate activation function. Functions such as sigmoid [26], TLU [27] (threshold linear unit), and Gaussian [28] are used, but the most common are rectified linear unit (ReLU) [29] or TanH [22].

## 3. Materials and Methods

In our experiments, we compared how shallow and deep learning methods cope with detecting hidden data in JPEG images. We tested a variety of deep and shallow ML-based classifiers and various feature spaces. Initially, we used raw DCT coefficients as input for the tested methods. As it did not produce satisfactory results, we extracted various parameters from the images. We performed experiments in DCTR, GFR, and PHARM feature spaces. We taught our models features extracted from pairs of images: without and with steganographically hidden data. Details of the data and the classifiers used are presented in the next subsections.

### 3.1. Datasets Used

We used the "Break Our Steganograhic System" (BOSS) image collection [30], which contains 10,000 black and white photos (with no hidden data). The photos were converted into JPEG with a quality factor of 75. Then, we generated three other sets of images, hiding random data with a density of either 0.4 or 0.1 bpnzac, using three different steganographic algorithms: J-Uniward, nsF5, and UERD. We used their code published at [31]. All experiments, including generation of the steganographic files, were run on a virtual machine with 64 GB RAM and 8 vCPU cores of Intel Xeon Gold 5220 processor, running on a DELL PowerEdge R740 server. Each dataset was divided in parallel into training and test subsets, in the ratio of 90:10.

### 3.2. Configuration of Ensemble Classifier

The base component of the shallow classifier is the ensemble classifier based on the FLD model [21]. A diagram presenting the way the ensemble classifier operates is shown in Figure 2. The set of feature vectors created by extracting DCTR, GFR, or PHARM characteristics from pictures is used to generate smaller subsets through a random selection of samples from the original set (a process called bootstrapping). These subsets are then used to train individual base learners independently from each other to diversify their classification logic. Throughout the training process, the size of the subset and the population of the ensemble (the number of base learners) is adjusted to minimize the out-of-bag error of the system. These subsets are then used to train individual base learners. Upon testing, each base learner reaches its decision independently of others and the results from the whole "population" are aggregated to produce a single decision.

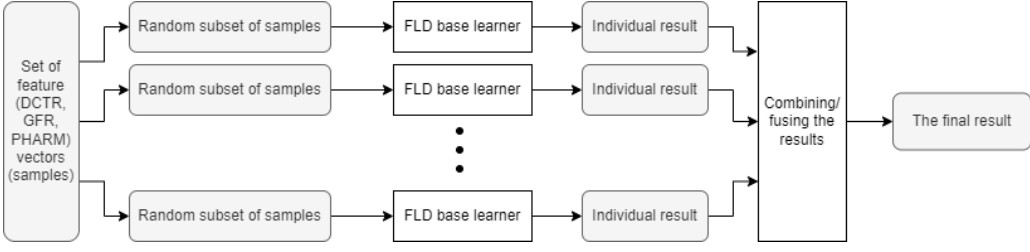

**Figure 2.** A diagram showing the structure of the ensemble classifier.

In our work, we focused on maximizing the detection ability of this classifier through the use of various methods to combine the votes of base learners. While the individual votes in the original ensemble were fused by simply choosing the more popular classification decision, we decided to explore the potential gain of employing machine learning for this. We trained the original ensemble classifier and then used it to obtain the decisions of all base learners for both the training and testing sets. The resulting data formed new feature vectors, which were used for further analysis with different ways of combining the votes of individual base learners. We performed this analysis using primarily methods implemented in the scikit-learn library [32]. As such, the original ensemble became a dimension-reducing layer.

### 3.3. Deep Learning Environment

The neural network environment was based on the Keras [33] and Tensorflow [34] library due to the simplicity of the model definition. The network architecture was mainly based on the Dense-BatchNormalization structure, but not using the convolution part as described in the available literature. We also tested various activation functions for the dense layer, such as sigmoid, softsign, TanH, and softmax, but the best results were obtained for the ReLU function. We used two optimizers: adaptive moment estimation (Adam) [35] and stochastic gradient descent (SGD) [36], which gave different results depending on the type of input parameters. The last parameter that significantly influenced the model learning efficiency was the learning rate. We found that lowering it gave very promising results without changing the network architecture and the optimizer. One of the network configurations used is displayed in Figure 3.

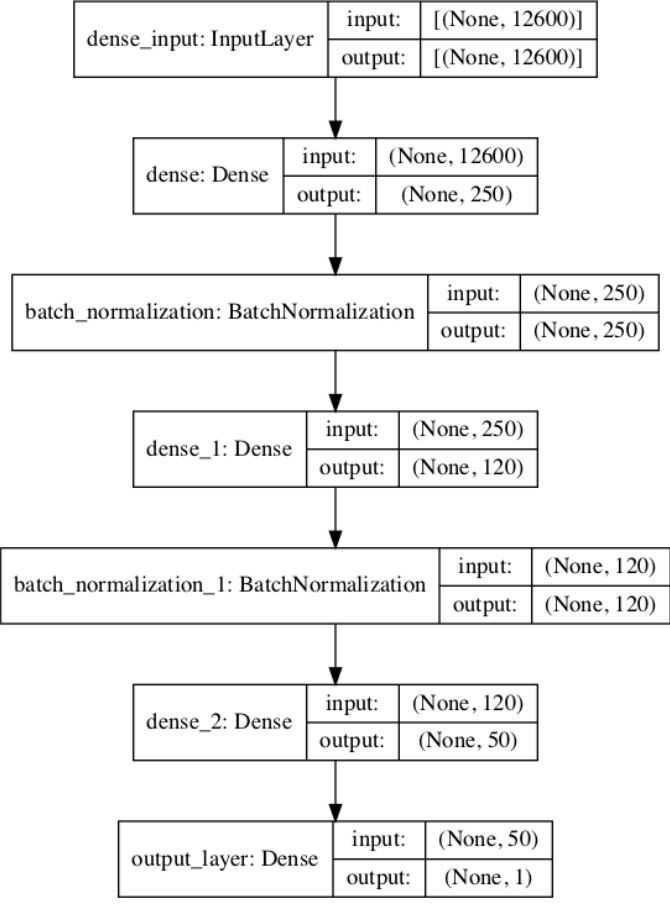

**Figure 3.** Example of 3 Dense-BatchNormalization neural networks used for detecting data hidden by a JPEG-based steganographic method.

### 3.4. Testing Scenarios

For the shallow ML-based algorithms, we decided to focus on the ensemble classifier, which has been reported in related studies as one of the most promising. During our experiments, we verified a number of ML-based methods used to combine the set of votes coming from all base learners to return the final classifier decision. These include: linear regression, logistic regression, linear discriminant analysis (LDA), and *k* nearest neighbors (*k*-NN). Moreover, the majority voting scheme (i.e., choosing the most popular classification decision, which is the original ensemble vote fusion method), as well as unquantized majority voting (i.e., classification based on the sum of non-quantized decisions of the whole ensemble) was included for comparison.

As for deep learning methods, two network architectures were selected for experiments:

- Three dense layers with the ReLU activation function, with 250 neurons in the first, 120 in the second, and 50 in the third, used in four reference models;
- Two dense layers also with the ReLU function, having 500 neurons in the first layer and 250 in the second, used in the last (fifth) reference model.

We decided not to use any convolutional layers due to their high computational requirements. However, we used additional normalization layers (BatchNormalization) between the dense layers. Half of the three-layer dense models used the Adam optimizer and half used SGD models, while the two-layer dense model used only the Adam optimizer. In the case of learning rate for the Adam optimizer, the values $1 \times e^{-4}$ or $1 \times e^{-5}$ were used, while for SGD, $1 \times e^{-3}$ or $1 \times e^{-4}$ were used. The version of the SGD optimizer with learning rate $1 \times e^{-3}$ or $1 \times e^{-4}$ and the $1 \times e^{-4}$ version of the Adam optimizer were omitted here, because they yielded much worse results compared to the version with three dense layers. In total, five different neural network configurations were tested for steganography detection.

### 3.5. Evaluation Metrics

To evaluate the models created, we employed commonly used metrics. The first is accuracy, which indicates what percentage of the entire set of classified data is the correct classification. The second metric is precision, which determines what proportion of the results indicated by the classifier as belonging to a given class actually belongs to it. Another metric is recall, which determines what part of the classification results of a given class is detected by the model. The fourth metric analyzed is the F1-score, which is the harmonic mean of precision and recall. It reaches 1.0 when both components give maximum results. The last metric we used to test the effectiveness of the model is the area under the ROC curve (AUC). We will also present the ROC curves themselves, as they visually present the effectiveness of the detection model.

In our results, we focus on evaluating the accuracy for each model combination, while for the best parameters we also provide the values of the other metrics. Since the testset is ideally balanced, the accuracy score is not biased and reflects well the detection ability of a given classifier.

## 4. Results

Tables 1 and 2 show the results of steganography detection obtained by shallow and deep methods, respectively. We display the accuracy values achieved for various steganographic algorithms and various hidden data densities, accompanied by average accuracies for each classifier/parameter combination.

**Table 1.** Accuracy of image steganography detection (in percentages) for various classifiers and ensemble configurations. The best values in each column are shown in bold.

| Classifier | Parameters | J-Uniward | | nsF5 | | UERD | | Avg. |
|---|---|---|---|---|---|---|---|---|
| | | 0.1 | 0.4 | 0.1 | 0.4 | 0.1 | 0.4 | |
| Majority voting | DCTR | 50.9 | 84.9 | 78.7 | **99.9** | 66.1 | 95.3 | 79.3 |
| | GFR | 54.9 | 89.4 | 70.4 | 99.2 | 65.5 | **95.9** | 79.2 |
| | PHARM | 53.9 | 84.9 | 70.5 | 98.7 | 64.5 | 94.6 | 77.9 |
| Unquant. majority voting | DCTR | 53.3 | 85.0 | 79.4 | **99.9** | 66.3 | 95.3 | 79.9 |
| | GFR | 55.4 | 89.5 | 70.3 | 99.2 | 65.9 | 95.7 | 79.3 |
| | PHARM | 54.1 | 50.8 | 50.7 | 51.7 | 50.0 | 50.0 | 51.2 |
| Linear regression | DCTR | 54.2 | 85.6 | **79.7** | **99.9** | 66.1 | 95.2 | **80.1** |
| | GFR | **56.3** | **89.8** | 70.2 | 99.2 | 66.2 | 95.7 | 79.6 |
| | PHARM | 54.5 | 85.7 | 70.1 | 98.8 | 64.0 | 94.0 | 77.9 |
| Logistic regression | DCTR | 54.3 | 85.4 | 79.4 | 99.7 | 66.2 | 94.9 | 79.9 |
| | GFR | **56.3** | 89.5 | 70.1 | 99.1 | 65.9 | 95.5 | 79.4 |
| | PHARM | 54.6 | 62.0 | – | 98.5 | 64.5 | 94.7 | 70.7 |
| LDA | DCTR | 54.2 | 85.6 | **79.7** | **99.9** | 66.1 | 95.2 | **80.1** |
| | GFR | **56.3** | 89.7 | 70.2 | 99.1 | 66.2 | 95.7 | 79.5 |
| | PHARM | 54.4 | 85.7 | 70.1 | 98.8 | 64.0 | 94.0 | 77.8 |
| *k*-NN | DCTR | 53.8 | 85.0 | 78.9 | **99.9** | **66.8** | 95.2 | 79.9 |
| | GFR | 56.1 | **89.8** | 70.2 | 99.3 | 66.1 | **95.9** | 79.6 |
| | PHARM | 54.8 | – | – | 93.9 | 63.5 | 94.8 | 67.8 |

**Table 2.** Accuracy of image steganography detection (in percentages) for various architectures of neural networks and optimizers. The best values in each column are shown in bold.

| Network Arch. | Optimizer | Parameters | J-Uniward | | nsF5 | | UERD | | Avg. |
|---|---|---|---|---|---|---|---|---|---|
| | | | 0.1 | 0.4 | 0.1 | 0.4 | 0.1 | 0.4 | |
| 250 × BN × 120 × BN × 50 (3 layers) | Adam $1e^{-4}$ | DCTR | – | 83.1 | **76.3** | 98.8 | **66.5** | **94.5** | **78.3** |
| | | GFR | – | 86.5 | 68.3 | 95.5 | 63.4 | 92.9 | 76.1 |
| | | PHARM | – | 74.7 | 62.3 | 95.9 | 51.4 | 88.5 | 70.5 |
| | Adam $1e^{-5}$ | DCTR | – | 83.0 | 74.2 | **99.7** | 64.7 | 93.1 | 77.5 |
| | | GFR | – | **88.4** | 68.0 | 98.2 | 62.6 | 92.5 | 76.6 |
| | | PHARM | – | 76.1 | 66.1 | 93.4 | 55.5 | 89.4 | 71.8 |
| | SGD $1e^{-3}$ | DCTR | – | 77.0 | 73.8 | 99.6 | 62.8 | 91.4 | 75.8 |
| | | GFR | – | 78.6 | 68.8 | 97.5 | 58.5 | 91.9 | 74.2 |
| | | PHARM | – | 58.4 | 51.4 | 59.8 | 50.6 | 61.5 | 55.3 |
| | SGD $1e^{-4}$ | DCTR | – | 73.3 | 52.1 | 99.1 | 51.6 | 91.9 | 69.7 |
| | | GFR | – | 82.2 | 58.6 | 97.6 | 52.1 | 91.9 | 72.1 |
| | | PHARM | – | 60.9 | – | 69.7 | 50.6 | 68.9 | 58.4 |
| 500 × BN × 250 (2 layers) | Adam $1e^{-5}$ | DCTR | – | 80.8 | 73.5 | 99.6 | 61.9 | 93.5 | 76.6 |
| | | GFR | 53.6 | 86.4 | 67.6 | 97.4 | 64.2 | 91.9 | 76.9 |
| | | PHARM | – | 75.0 | 54.1 | 94.2 | 54.0 | 87.9 | 69.2 |

On average, the use of ML for ensemble vote combination allowed for higher detection accuracy when using the systems based on DCTR or GFR features, despite marginally worse performance in certain cases (such as GFR features extracted from nsF5-modified files at 0.1 bpnzac). The PHARM-features-based classifiers sometimes yielded results worse than when using the default, majority-based scheme, or failed to converge altogether. There was no combination of type of parameters used (DCTR, GFR, PHARM) and method of fusing base-learner votes into the final decision that outperformed the others in all testing scenarios. The configuration that, on average, achieved the best results for the

steganographic algorithms tested turned out to be the linear regression classifier fed with DCTR features. While using linear discriminant analysis (LDA) to fuse votes coming from a system operating on DCTR parameters achieved equal averaged accuracy, linear regression is considered in further sections due to slightly better performance with GFR and PHARM features.

As for the deep learning algorithms (Table 2), the lowest accuracy was obtained for the set based on J-Uniward. Better results in terms of accuracy were obtained for the sets based on UERD, and the best were achieved for nsF5. When analyzing the tested configurations, the worst results are those based on the SGD optimizer, while the configurations based on Adam performed better at higher learning rates. Comparing the configuration based on three layers and two layers, the results are rather similar for the Adam optimizer.

Looking at the various feature spaces, it can be seen that the least accurate results were always obtained for PHARM. On the other hand, the results obtained for the DCTR and GFR parameters for all combinations were much better and rather similar, which means that most probably they can be used interchangeably in JPEG steganalytic tools.

These observations are further confirmed in Figure 4. The PHARM parameters always yielded the worst results. The GFR features usually gave slightly better results for the higher embedding rate (0.4 bpnzac), while for the lower embedding rate (0.1 bpnzac) it was the DCTR feature space that turned out to be slightly better for most of the tested classifiers, both shallow and deep learning-based.

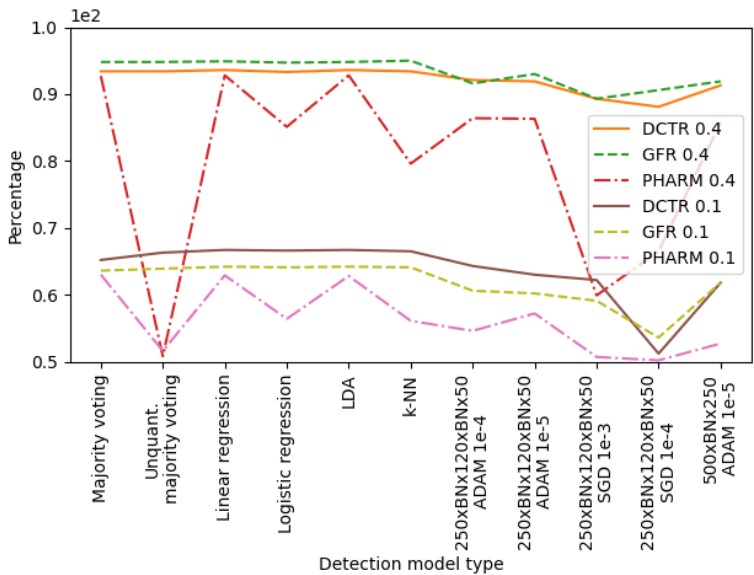

**Figure 4.** Accuracy achieved for various feature vectors against classifiers or network architectures.

After conducting the research, we selected the best configurations for specific types of sets, differentiating for shallow and deep learning methods, and calculated the remaining metrics. Their outcomes are visualized in Figure 5, while the details are shown in Tables 3 and 4. Based on Figure 6, one can notice that the differences between the main evaluation metrics for the best shallow and deep methods for density 0.4 bpnzac are only minor. A somewhat higher difference can be observed for all the tested steganographic algorithms applied at the lower embedding rate: 0.1 bpnzac. Here, the ensemble (shallow) classifier usually turned out to be slightly better.

**Table 3.** Results of image steganography detection (in percentages) for the best shallow method (linear regression).

| Metric | J-Uniward | | nsF5 | | UERD | | Avg. |
|---|---|---|---|---|---|---|---|
| | 0.1 | 0.4 | 0.1 | 0.4 | 0.1 | 0.4 | |
| Accuracy | 54.2 | 85.6 | 79.7 | 99.9 | 66.1 | 95.2 | 80.1 |
| Precision | 54.2 | 86.4 | 80.2 | 99.9 | 67.9 | 96.1 | 80.8 |
| Recall | 54.1 | 84.4 | 78.9 | 99.8 | 61.1 | 94.1 | 78.7 |
| F1-score | 54.1 | 85.4 | 79.5 | 99.9 | 64.3 | 95.1 | 79.7 |
| AUC | 54.9 | 91.8 | 87.7 | 99.9 | 72.4 | 98.8 | 84.3 |

**Table 4.** Results of image steganography detection (in percentages) for the best deep learning method ($250 \times BN \times 120 \times BN \times 50$ with Adam $1 \times e^{-4}$ based on DCTR parameters).

| Metric | J-Uniward | | nsF5 | | UERD | | Avg. |
|---|---|---|---|---|---|---|---|
| | 0.1 | 0.4 | 0.1 | 0.4 | 0.1 | 0.4 | |
| Accuracy | 50.2 | 83.1 | 76.3 | 98.8 | 66.5 | 94.5 | 78.2 |
| Precision | 50.2 | 80.2 | 74.5 | 98.5 | 63.6 | 94.6 | 76.9 |
| Recall | 46.1 | 87.7 | 80.7 | 99.0 | 78.7 | 94.3 | 81.1 |
| F1-score | 48.1 | 83.8 | 77.3 | 98.8 | 70.1 | 94.4 | 78.8 |
| AUC | 50.3 | 91.6 | 84.5 | 99.8 | 72.8 | 98.7 | 83.0 |

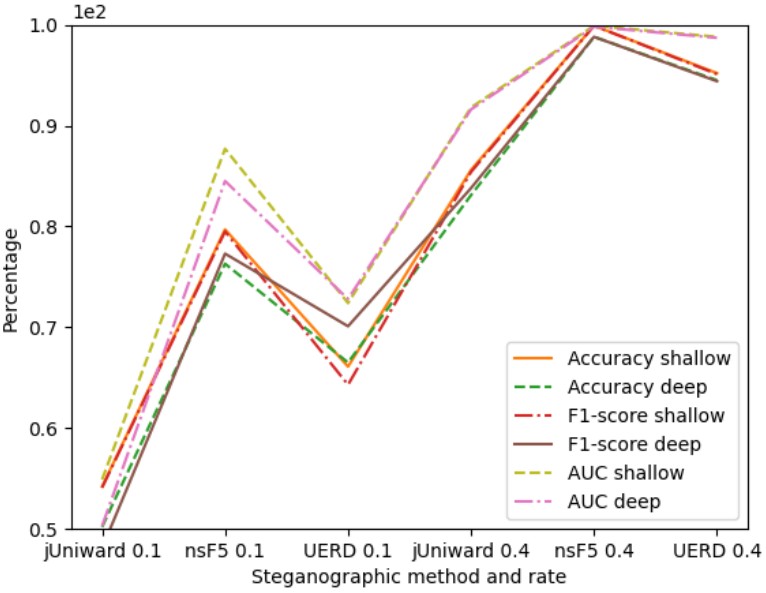

**Figure 5.** Visualization of evaluation results for the best shallow and deep steganalytic algorithms.

These observations are confirmed by the scores shown in Tables 3 and 4. The highest difference is for the J-Uniward 0.1 set, where the difference is about 4% relative, while for other sets we usually observe about 1–2% relative advantage in favor of the ensemble classifier, which means that these differences are only minor.

In total, the parameters of detecting data hidden using nsF5 at 0.4 embedding rate are close to 100%, regardless of the method. In contrast, the metrics for detection of data hidden with J-Uniward at 0.1 bpnzac are very poor. For the ensemble classifier with linear regression, all metrics are around 54%, while for the best neural network for most of the results are at the chance level. In general, the detection of all the tested JPEG-based steganographic methods working at the embedding rate of 0.4 bpnzac can be conducted with accuracy, with F1-score and AUC scores above 85%. The detection of hidden content embedded at a low rate of 0.1 bpnzac is problematic both for shallow and deep methods.

In the best case, the detection accuracy reached 85% for the easiest, nsF5 algorithm, while it was significantly lower for UERD and J-Uniward.

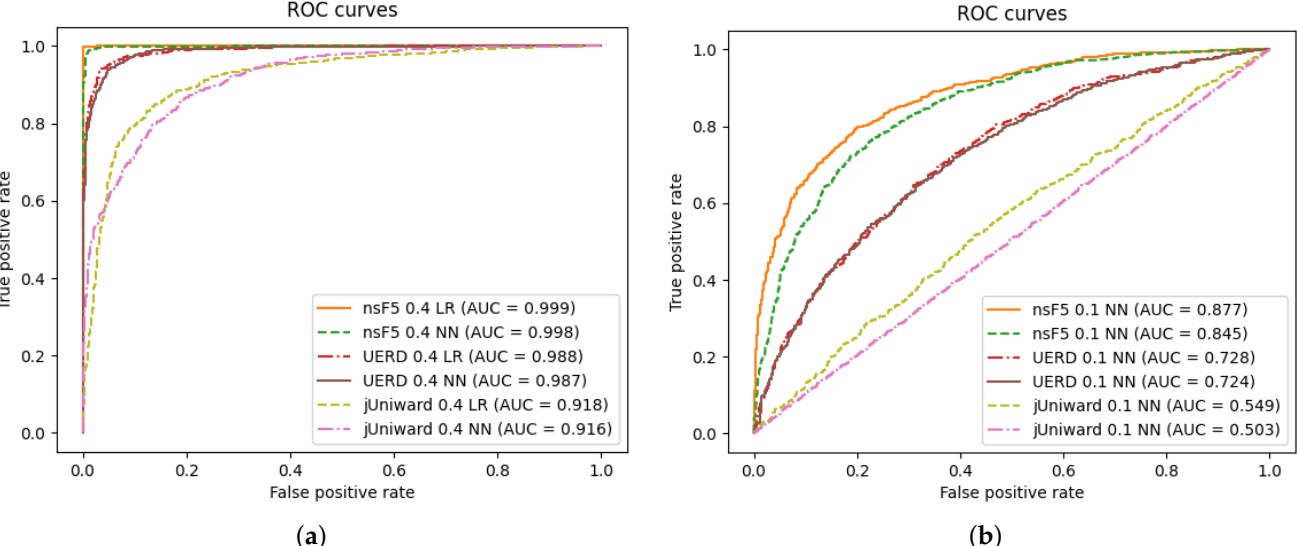

**Figure 6.** Comparison of ROC curves for the best neural network and the best ensemble classifier for data hidden (**a**) with density 0.4 bpnzac and (**b**) with density 0.1 bpnzac.

## 5. Discussion

The results obtained were compared to the results of similar studies. In article [37], the research was also carried out for the BOSS dataset using an alternative version of J-Uniward—an SUniward algorithm. The model was based on the Convolution-BatchNormalization Dense neural network scheme. The authors obtained AUC at the level of 97.9% for density 0.4 bpnzac. It was a similar result to our best model, which is much less computationally complex, due to the lack of a convolutional layer.

Articles [13,14] also conducted research on the BOSS dataset using the DCTR parameters and the Gabor filters on the J-Uniward algorithm, while using different decision models. They obtained a detection error, calculated based on false-alarm and missed-detection probabilities, as proposed in [37], of around 0.04–0.05 for DCTR, and the out-of-bag error for the Gabor filter was around 0.39, both assessed for density 0.4. Unfortunately, these results are difficult to compare with ours due to the different metrics used and the testing methodology.

During experimentation with various types of neural network layers, we noticed that adding a normalization layer significantly improved the effectiveness of a model. For example, for the nsF5 method, it improved the results by about 15–20% relative, while for the J-Uniward 0.4 case it made it possible to build a reasonable model. Without this layer, the network tended to classify all images into one class. It indicates that normalization layers in the Convolution-Dense-BatchNormalization model are indispensable, in contrast to convolution layers, the lack of which can be compensated for by, for example, choosing a different feature space.

The results achieved in our study for the three-layer and two-layer network configurations were quite similar for the Adam optimizer. This may indicate that enlarging the architecture of a neural network is pointless, as it can only have a negative impact on the computational efficiency of the neural model.

It is noteworthy that the BOSS dataset used in this study is comprised exclusively of grayscale images. Thus, only the luma channel was present in each JPEG file. However, the steganographic algorithms tested (nsF5, J-Uniward, and UERD) typically introduce changes only to DCT coefficient values of the luma channel. As such, non-grayscale (colored) images can easily be analyzed in the same way as the files from the BOSS dataset, with the chrominance channels being ignored in the detection process.

### 6. Conclusions and Future Work

In this article, we analyzed the effectiveness of detecting hidden content in JPEG images using either shallow, ensemble classifiers, or deep learning methods. We found that performance depended heavily on the steganographic method used and on the density of the embedded hidden data. While detecting the presence of content hidden with the nsF5 algorithm at the density 0.4 bpnzac is almost perfect, the detection of data hidden using J-Uniward at 0.1 bpnzac is hardly possible, regardless of the analysis method used.

One of the aims of our study was to find the best feature space for image steganalysis. DCTR and GFR parameters yielded the best results, while the feature space built on the PHARM parameters returned worse scores. Therefore, we recommend extracting either DCTR or GFR features when scanning JPEG files for security purposes, e.g., by antimalware software.

We also found that the performance of the best deep learning algorithm (with the network architecture: $250 \times BN \times 120 \times BN \times 50$ and the Adam $1 \times e^{-4}$ optimizer) was either similar or slightly inferior to that of the best ensemble classifier built on linear regression. Therefore, we claim that carefully selected ensemble classifiers could be a promising alternative to deep learning methods in the field of image steganalysis.

Future work could concentrate on searching for effective detection methods for rates of embedding hidden data lower than 0.4 bpnzac, bearing in mind malware or advanced persistent threats (APTs) exchanging lower amounts of data. Researchers should especially focus on steganalysis of algorithms such as J-Uniward, which turned out to be particularly difficult to detect. It would be also interesting to see an application of elaborated algorithms, e.g., in an intrusion detection system (IDS). A study on the impact of characteristics of the hidden data (random, text, script) on the detectability of a JPEG-based steganographic method would also be beneficial.

**Author Contributions:** Conceptualization, M.P., K.S. and A.J.; methodology, M.P., M.K. and A.J.; software, M.P. and M.K.; validation, M.P. and M.K.; investigation, M.P., M.K., K.S. and A.J.; data curation, M.P.; writing—original draft preparation, M.P., M.K. and A.J.; writing—review and editing, M.P., M.K., K.S. and A.J.; visualization, M.P.; supervision, A.J. and K.S.; project administration, A.J.; funding acquisition, A.J. and K.S. All authors have read and agreed to the published version of the manuscript.

**Funding:** The study has been supported by the SIMARGL Project—Secure Intelligent Methods for Advanced Recognition of malware and stegomalware, with the support of the European Commission and the Horizon 2020 Program, under Grant Agreement No. 833042.

**Data Availability Statement:** Image data are freely available at https://www.kaggle.com/datasets/h2020simargl/jpeg-stegochecker-dataset (accessed on 10 May 2022).

**Conflicts of Interest:** The authors declare no conflict of interest.

**Acknowledgments:** The authors wish to thank the creators of the BOSS dataset: Tomáš Pevný, Tomáš Filler and Patrick Bas, for creating and publishing their data.

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
