# Peer review of "Detection of Image Steganography Using Deep Learning and Ensemble Classifiers"

_electronics, doi:10.3390/electronics11101565_

Round 1

Reviewer 1 Report

In the current work, the problem of detecting JPEG images which have been steganographically manipulated is fully discussed. The performance of employing various shallow and deep learning algorithms in image steganography detection is analyzed. 

The article is interesting and merits publication in Electronics after taking into consideration the following minor corrections:

  1. The introduction part should be extended to cover the cons/pros of the work.
  2. Why the study is focused on JPEG-based images? why not jpg?
  3. Elaborate more on the structure of the ensemble classifier.
  4. Mention the specification of the machine used to depict the figures, as well as the platform used to produce these figures.
  5. Comment on future extensions in this direction.
  6. Proofreading is required.

Reviewer 2 Report

This study is devoted to the issues of detecting the facts of hidden data transmission by embedding malicious software in jpeg image files. The research was based on the analysis and evaluation of machine learning models and approaches to determine the main steganographic algorithms to integrate hidden data into jpeg image files. Despite the sufficient representation of scientific results in the field of this issue, the goal and objectives of this article are relevant and are in a large demand at this time. The authors conducted a detailed review of the background of the problem of detecting steganographic processing of image data files, analyzed, evaluated, and proposed machine learning approaches to determine the facts of covert data transmission. Including the models of the author that demonstrate promising results is considered. In general, the study is well structured, the materials are presented in an accessible and consistent form, and the basic minimum for understanding the problem under study is given. In addition, among the main results of the study obtained by the authors, it is worth highlighting a deep analysis and search for the most effective feature space to detect the integration of malware into the image data structure to hide the fact of transmission. In the process of getting acquainted with the results of the study, several questions / suggestions arose:

  1. (row 169-175) As the initial data set for the investigation, the authors chose the collection Break Our Steganograhic System, based on which the training and test data sets were generated. The data generation procedure itself can be described in more detail. Despite the presence of all data in the public repository of projects in the text of the article, specific illustrations and examples can be given.
  2. In row 159 of the study, information is provided on a comparative analysis of activation functions for a dense layer, as a result of which ReLu was chosen. I would like to understand in more detail what activation functions were considered and how it affected the final quality metrics of the models.
  3. The possibility of adding (where applicable) heatmaps for a visual representation of precision, recall, and f1 score metrics may be considered.
  4. In the Introduction, the authors talk about the potential integration of their fundings with anti-malware software. However, further in the text of the study, no assessment of possible practical integration is provided. This remark is not important, but it would be interesting to learn about the possible experience of the authors in this direction.

The article is logically presented and meets the criteria of scientific novelty. The minimum of materials necessary to evaluate the results of the study is given. There are certain doubts about the compliance of the subject of the research with the goals and objectives of the journal. However, the study is recommended for publication with minor revisions.
